# Ag_3_PO_4_-Deposited TiO_2_@Ti_3_C_2_ Petals for Highly Efficient Photodecomposition of Various Organic Dyes under Solar Light

**DOI:** 10.3390/nano12142464

**Published:** 2022-07-18

**Authors:** Ngoc Tuyet Anh Nguyen, Hansang Kim

**Affiliations:** Department of Mechanical Engineering, Gachon University, Seongnam 13120, Korea; anh2510@gachon.ac.kr

**Keywords:** Ti_3_C_2_ MXene, Ag_3_PO_4_, photocatalyst, organic dyes, solar light, Ag_3_PO_4_/TiO_2_@Ti_3_C_2_

## Abstract

Two-dimensional Ti_3_C_2_ MXenes can be used to fabricate hierarchical TiO_2_ nanostructures that are potential photocatalysts. In this study, the photodecomposition of organic dyes under solar light was investigated using flower-like TiO_2_@Ti_3_C_2_, deposited using narrow bandgap Ag_3_PO_4_. The surface morphology, crystalline structure, surface states, and optical bandgap properties were determined using scanning electron microscopy (SEM), transmission electron microscopy (TEM), X-ray diffraction (XRD), X-ray photoelectron spectroscopy (XPS), nitrogen adsorption analysis, and UV-Vis diffuse reflectance spectroscopy (UV-DRS). Overall, Ag_3_PO_4_-deposited TiO_2_@Ti_3_C_2_, referred to as Ag_3_PO_4_/TiO_2_@Ti_3_C_2_, demonstrated the best photocatalytic performance among the as-prepared samples, including TiO_2_@Ti_3_C_2_, pristine Ag_3_PO_4_, and Ag_3_PO_4_/TiO_2_ P25. Organic dyes, such as rhodamine B (RhB), methylene blue (MB), crystal violet (CV), and methylene orange (MO), were efficiently degraded by Ag_3_PO_4_/TiO_2_@Ti_3_C_2_. The significant enhancement of photocatalysis by solar light irradiation was attributed to the efficient deposition of Ag_3_PO_4_ nanoparticles on flower-like TiO_2_@Ti_3_C_2_ with the efficient separation of photogenerated e-/h+ pairs, high surface area, and extended visible-light absorption. Additionally, the small size of Ag_3_PO_4_ deposition (ca. 4–10 nm diameter) reduces the distance between the core and the surface of the composite, which inhibits the recombination of photogenerated charge carriers. Free radical trapping tests were performed, and a photocatalytic mechanism was proposed to explain the synergistic photocatalysis of Ag_3_PO_4_/TiO_2_@Ti_3_C_2_ under solar light.

## 1. Introduction

Globally, as industrialization and urbanization accelerate, the demand for food and consumer goods (clothing, electronics, furniture, vehicles, etc.) increases. Unsustainable resource use and inefficient waste management, however, contribute to environmental problems, which have become a source of concern [1,2,3]. Numerous harmful pollutants endanger the environment and human health, such as heavy metal ions, toxic gases, and organic dye [4,5]. The majority of organic dyes found in wastewater originate from various industrial waste streams, including paint, dyeing clothing, bleaching paper, synthesizing rubber, and processing plastic, thereby, resulting in water pollution.

Organic dyes at high concentrations (5–1500 mg/L) cause significant environmental and organismal toxicity, including carcinogenic, mutagenic, and teratogenic effects [6,7,8]. Owing to their low removal capacity, conventional biological processes used to remove dyes, such as flocculation, filtration, precipitation, and coagulation, have gradually become obsolete [9,10]. In comparison, heterogeneous photocatalysts used in advanced oxidation processes are considered promising alternatives for the removal of a wide variety of organic pollutants with a highly efficient degradation rate [11].

Titania (TiO_2_) is the most widely used and recognized photocatalytic material in environmental remediation owing to its superior photocatalytic performance, non-toxicity, and stability [12,13,14]. However, pure TiO_2_ has a low quantum efficiency and a large bandgap, limiting its application (i.e., electrons can be triggered only under UV light). In addition, the rapid recombination of photogenerated electron–hole pairs reduces its degradation efficiency [15]. Recently, several strategies have been proposed to solve these problems by fabricating TiO_2_ derived from Ti_3_C_2_ [16].

Despite the absence of noble metal deposition (Au, Ag, Pt, Ru_2_O, etc.), the flower-shaped TiO_2_@Ti_3_C_2_ composite exhibits significantly enhanced photocatalytic activity [4,17]. Recent work has been published on the synthesis of TiO_2_@Ti_3_C_2_ nanoflowers from the MAX phase (Ti_3_AlC_2_) [4,16,17]. They demonstrated that the presence of Ti_3_C_2_ in the TiO_2_ structure eliminates electron–hole pair recombination, maximizes charging transfer, and expands the light absorption area to use visible light. Under solar light irradiation, the degradation efficiencies of RhB were 97% within 40 min and 95% within 60 min. Therefore, flower-shaped TiO_2_@Ti_3_C_2_ is a highly promising photocatalyst [4,16,17].

Silver-based semiconductor photocatalysts have been widely used due to their high efficiency as visible-light-driven photocatalysts [18]. Among silver-based photocatalysts, trisilver phosphate (Ag_3_PO_4_) with a narrow bandgap (2.36 eV) has attracted the photocatalytic field’s attention due to its high oxidation capacity and ability to remove pollutants under visible light illumination [18,19,20,21]. Nevertheless, Ag_3_PO_4_ continues to fall short of meeting the demand for large-scale industrial applications, primarily due to its low reuse rates, as it suffers greatly from severe photocorrosion [22,23].

This is due to the intrinsic fast charge recombination, self-corrosion, and photocorrosion of silver or silver oxides in the absence of a sacrificial reagent. Therefore, the key strategy is to prevent its photocorrosion in practical applications. In this regard, heterostructure photocatalysts composed of Ag_3_PO_4_ and TiO_2_ semiconductors have attracted attention over the years due to their increased photocatalytic efficiency, increased stability, and lower noble metal consumption [20]. To our knowledge, no research on the combination of Ag_3_PO_4_ and Ti_3_C_2_-derived TiO_2_ has been conducted.

Herein, an Ag_3_PO_4_-deposited TiO_2_@Ti_3_C_2_ composite, Ag_3_PO_4_/TiO_2_@Ti_3_C_2_, was developed as a highly efficient visible photocatalyst. Flower-like TiO_2_ was initially synthesized from Ti_3_C_2_ MXenes through three consecutive steps of hydrothermal oxidation, ion exchange, and heating processes [4,16,17]. Thereafter, using an in-situ precipitation method, Ag_3_PO_4_ nanoparticles (NPs) were deposited on flower-like TiO_2_@Ti_3_C_2_. It is conceivable that the presence of metal-like Ti_3_C_2_ in the TiO_2_@Ti_3_C_2_ composite acts as an “electron sink”, facilitating the highly efficient photodegradation of organic dyes.

In addition, the flower-like morphology has a high surface area and an efficient deposition of Ag_3_PO_4_ on TiO_2_@Ti_3_C_2_ petals, which effectively separates electron–hole pairs and improves the solar light harvesting capability. Ag_3_PO_4_-deposited TiO_2_@Ti_3_C_2_ exhibited significant photocatalytic performance in the photodecomposition of a variety of organic dyes (including RhB, MB, CV, and MO).

## 2. Materials and Methods

### 2.1. Chemical and Materials

Titanium carbide powder (Ti_3_C_2_ MXenes) was provided by Invisible Co. Ltd., Seoul, Korea. Sodium hydroxide (NaOH) was purchased from Daejung Chemicals & Metals Co., Ltd., Siheung, Korea. Silver nitrate (AgNO_3,_ 99.9%) was obtained from Duksan Pure Chemicals Co., Ltd., Ansan, Korea. Hydrogen peroxide (H_2_O_2,_ 30%), hydrochloric acid (HCl), ethyl alcohol (C_2_H_5_OH), sodium phosphate (Na_3_PO_4_), methylene blue (MB), rhodamine B (RhB), methylene orange (MO), and crystal violet (CV) were purchased from Sigma-Aldrich, Munich, Germany. All chemicals were used directly, without any further treatment.

### 2.2. Preparation of TiO_2_@Ti_3_C_2_ Heterostructure

The procedure for fabricating flower-like TiO_2_@Ti_3_C_2_ was previously described by Vu Thi Quyen et al. [17]. First, 100 mg of Ti_3_C_2_ MXene was added and vigorously stirred for 15 min with a solution of 2 M NaOH (80 mL) and 30 wt% H_2_O_2_ (8 mL). Second, the hydrothermal reaction was performed by transferring the mixture into two 50 mL autoclave systems at 140 °C for 15 h. The dispersion was then allowed to cool naturally to room temperature, and the samples were washed with deionized (DI) water and ethanol several times before being dried in an oven at 60 °C for 12 h. The dried sample was soaked in a 0.05 M HCl solution (500 mL) for 12 h to ensure ion exchange. After heating the sample in a furnace at 500 °C for 5 h, the TiO_2_@Ti_3_C_2_ composite was obtained. The formation of TiO_2_ is presented by the following equations:2H_2_O_2_ → 2H_2_O + O_2_(1)
Ti_3_C_2_ + 5O_2_ → 3TiO_2_ + 2CO_2_(2)
3TiO_2_ + 2NaOH → Na_2_Ti_3_O_7_ + H_2_O(3)
Na_2_Ti_3_O_7_ + HCl → H_2_Ti_3_O_7_ + 2NaCl(4)
H_2_Ti_3_O_7_ → 3TiO_2_ + H_2_O(5)

### 2.3. Preparation of Ag_3_PO_4_ Particles

To prepare Ag_3_PO_4_ particles, a simple method was followed. To generate Ag_3_PO_4_, 0.1 M Na_3_PO_4_ (8 mL) was added dropwise to 0.1 M AgNO_3_ solution (24 mL) and stirred for 5 h in the dark. To remove redundant ions, the precipitate was centrifuged and washed with deionized water. The purified sample was dried overnight at 60 °C to obtain the Ag_3_PO_4_ powder.

### 2.4. Preparation of Ag_3_PO_4_/TiO_2_@Ti_3_C_2_

Ag_3_PO_4_/TiO_2_@Ti_3_C_2_ composites were prepared by adding various amounts of AgNO_3_ (x) and Na_3_PO_4_ (y) solutions with a ratio of x:y = 3:1. First, 0.05 g TiO_2_@Ti_3_C_2_ was added to 0.1 M AgNO_3_ solutions. The mixed solutions were vigorously stirred and sonicated for 10 min. Thereafter, 0.1 M Na_3_PO_4_ was gradually dropped into the solution and stirred for 5 h to form Ag_3_PO_4_. To avoid photocorrosion, the reaction was conducted in the dark. The resulting powder products were washed, centrifuged with DI water, and dried at 60 °C to obtain Ag_3_PO_4_/TiO_2_@Ti_3_C_2_. The powder obtained from mixing quantities x = 1, 2, 4, 6, and 8 mL AgNO_3_ with y = 0.33, 0.67, 1.33, 2, and 2.67 mL Na_3_PO_4_ corresponds to samples marked A1 to A5, respectively). The samples were kept in the dark throughout the preparation process to avoid photocorrosion. In comparison with sample A4, Ag_3_PO_4_/TiO_2_ P25 was also prepared under the same conditions.

The fabrication procedure for the composites is depicted in Figure 1. First, Ti_3_C_2_ MXene was subjected to hydrothermal oxidation, ion exchange, and heat treatment steps, transforming its accordion-like structure into a flower shape. Thereafter, Ag_3_PO_4_ NPs containing varying amounts of Ag_3_PO_4_ were deposited by in-situ precipitation on the flower-like structure to form the Ag_3_PO_4_/TiO_2_@Ti_3_C_2_ composites. Photodegradation tests were conducted under solar-driven light.

### 2.5. Photocatalysis of Dye Degradation

A Xe lamp (1000 W) was used as an artificial solar light source with a light intensity of 1000 mW/cm^2^. Under simulated solar light, the photocatalytic activities of the as-prepared samples were evaluated for the photodecomposition of RhB dye. Typically, 10 mg of each sample was added to 20 mL of an aqueous solution containing RhB (9 mg/L). Prior to light irradiation, the adsorption experiments of Ag_3_PO_4_, TiO_2_, and Ag_3_PO_4_/TiO_2_@Ti_3_C_2_ were performed by stirring the solutions in the dark for 15 min to improve dispersion and adsorption–desorption equilibrium. Thereafter, the solution was exposed to solar light. Following that, an aliquot of each sample (1 mL) was removed and centrifuged at the indicated intervals to obtain the supernatant for UV-vis spectrophotometer evaluation.

The degradation efficiency of the dye materials can be described by the following equation: degradation percentage (%) = C_o_ − C_t_/C_o_ × 100. To determine the rate constant of RhB degradation, the degradation kinetics were assumed to follow the pseudo-first order model [21,24]:
$$-ln\frac{C}{C_{0}}=kt$$

where C_o_ and C_t_ denote the initial concentration of RhB and a specific concentration of RhB after exposure to light for t minutes, respectively. Here, “*k*” is the pseudo-first-order rate constant calculated from the linear slope of ln(C_0_/C) versus t (time). 10 mg/L concentration of MB, MO, and CV were used for photodegradation with the same procedure. The absorbance changes for each dye were determined using a UV-Vis spectrometer at different wavelengths, RhB (554 nm), MB (664–665 nm), MO (464 nm), and CV (590 nm).

### 2.6. Characterization

The X-ray diffraction (XRD) patterns of the samples were determined using a Rigaku Smartlab X-ray diffractometer with Cu-Kα radiation (*λ* = 1.544 Å) (Rigaku Corporation, Tokyo, Japan). The morphologies and microstructures of the samples were investigated using a Hitachi S-4700 field emission scanning electron microscope (FE-SEM) (Hitachi Ltd., Tokyo, Japan) and FEI Tecnai transmission electron microscopy (TEM) (FEI, Hillsboro, OR, United States). XPS measurements were conducted using an X-ray photoelectron spectrometer (XPS, Multilab 2000, Thermo Scientific, Waltham, MA, United States). The surface area, pore size, and pore volume were measured using an N_2_ adsorption–desorption apparatus (ASAP 2020, Micromeritics Instrument Corp, Norcross, GA, USA). Optical properties of the samples were determined using a Jasco V770 UV-Vis diffuse reflectance spectrophotometer (Jasco Inc., Easton, MD, USA)

## 3. Results and Discussion

### 3.1. XRD Analysis

The diffraction patterns of the TiO_2_@Ti_3_C_2_, Ag_3_PO_4_, and A4 samples are shown in Figure 1a. The XRD data of Ag_3_PO_4_ revealed sharp and narrow dominant peaks at 33.3° (210) and 36.6° (211), which corresponded to the body-centered crystal structure and high crystallinity (JCPDS no. 06-0505) [25]. Furthermore, TiO_2_@Ti_3_C_2_ exhibited a distinct peak at 2θ = 25.3°, indicating the high crystallinity of the anatase phase with other weaker peaks at 37.8°, 53.9°, 55.34°, and 62.6° (JCPDS card No. 21-1272) [21]. A small peak at 48.5° confirmed the presence of Ti_3_C_2_ in the sample [4,16,17].

In the case of the composites, Figure 1b shows the diffraction patterns of Ag_3_PO_4_/TiO_2_@Ti_3_C_2_ (samples A3, A4, and A5), which clearly display both Ag_3_PO_4_ crystal and TiO_2_ anatase phase peaks, confirming that the composites were composed of Ag_3_PO_4_ and TiO_2_. The diffraction peaks of TiO_2_ in the heterostructure composite are weaker than those of Ag_3_PO_4_ owing to the lower crystallinity of TiO_2_. However, samples A1 and A2 containing a lower concentration of Ag_3_PO_4_ mostly exhibit peaks of TiO_2_ anatase at = 25.3° (JCPDS card No. 21-1272), instead of Ag_3_PO_4_ peaks due to self-corrosion [22].

### 3.2. Morpholoy Analysis by SEM, TEM

The accordion-like shape of Ti_3_C_2_ is shown in the SEM image in Figure 2a. After oxidization, ion exchange, and heat treatment, accordion-shape Ti_3_C_2_ transforms into TiO_2_@Ti_3_C_2_ with a flower shape (Figure 2b). Figure 2c depicts pure Ag_3_PO_4_ NPs with an irregular shape and a diameter of ca. 300–500 nm [19]. The SEM image in Figure 2d shows aggregated Ag_3_PO_4_-deposited TiO_2_@Ti_3_C_2_ that retains its flower-like shape. To overcome certain limitations of SEM analysis, which focuses on the surface morphology of samples, TEM and HRTEM images were used to clarify the transmission morphology and crystalline structure of Ag_3_PO_4_/TiO_2_@Ti_3_C_2_ (sample A4). According to the EDX spectrum (Appendix A), the main elements present in the sample are carbon, oxygen, titanium, silver, and phosphorous. C, O, and Ti were obtained from TiO_2_@Ti_3_C_2_, with a portion of the O derived from Ag_3_PO_4_. It can be demonstrated that Ag_3_PO_4_ is coated on the surface of TiO_2_@Ti_3_C_2_.

Figure 3a shows a typical TEM image of the flower-shaped TiO_2_@Ti_3_C_2_ with some petal fragments and large agglomerated Ag_3_PO_4_ on it. However, the higher magnification images in Figure 3b,c demonstrate that the dominant Ag_3_PO_4_ NPs were formed and deposited on each TiO_2_@Ti_3_C_2_ nanorod. Positively charged silver ions were attracted to the surface of negatively charged TiO_2_, and the reaction between Ag^+^ and PO_4_^3−^ occurred immediately upon the addition of Na_3_PO_4_ solution, resulting in the deposition of Ag_3_PO_4_ NPs on TiO_2_@Ti_3_C_2_ rods [26]. The TEM images show that precipitating Ag_3_PO_4_ using TiO_2_@Ti_3_C_2_ nanorods as a template can reduce particle aggregation owing to the high specific surface area of the nanorods, which affects Ag_3_PO_4_ nucleation and reduces the diameter to around 4–10 nm.

Additionally, the Ag_3_PO_4_ nanoparticles were relatively stable and did not detach during sonication as part of the TEM preparation process, whereas the agglomerated Ag_3_PO_4_ fragments moved and changed position during real-time TEM measurements. Figure 3c,d show the high-resolution TEM (HR-TEM) images of the Ag_3_PO_4_/TiO_2_@Ti_3_C_2_ composite. TiO_2_@Ti_3_C_2_ (101) has lattice fringes with an interplanar spacing of 0.35 nm, while cubic Ag_3_PO_4_ (210) has an interplanar spacing of 0.26 nm (Figure 3d). A SAED image further demonstrates the clear formation and polycrystallinity of these materials, which is shown in Appendix A. These TEM images can assist in resolving the initial ambiguity regarding the morphology of Ag_3_PO_4_/TiO_2_@Ti_3_C_2_ shown in the SEM images.

Figure 4 depicts the scanning TEM (STEM) image of Ag_3_PO_4_/TiO_2_@Ti_3_C_2_ with elemental mapping. The high-angle annular dark-field (HAADF) STEM image of Ag_3_PO_4_/TiO_2_@Ti_3_C_2_ shows areas with different contrasts, with the brighter regions representing Ag_3_PO_4_ particles and the darker regions representing TiO_2_@Ti_3_C_2_ rods. The elemental mapping analysis displays C, O, P, K, and Ag elements from an arbitrary area. In detail, the Ti signal is primarily associated with the backbone of TiO_2_@Ti_3_C_2_, whereas the P and Ag signals are associated with the large agglomerated Ag_3_PO_4_ fragments on top of the flower and Ag_3_PO_4_ NPs decorated alongside the petals. Additionally, the O signals are distributed uniformly throughout the composite. The elemental mapping results provide more comprehensive and clear observations of the elemental distribution across the whole composite, corresponding to the SEM and HR-TEM above.

### 3.3. XPS Results

To characterize the chemical compositions and elemental states of the as-prepared samples, XPS measurements were conducted. The survey spectra of Ti_3_C_2_ MXene, TiO_2_@Ti_3_C_2_ flowers, and Ag_3_PO_4_/TiO_2_@Ti_3_C_2_ illustrate the characteristic elemental peaks at approximately 532.35, 455.16, and 284.96 eV allocated to O 1s, Ti 2p, and C 1s, respectively (Appendix A). The weak peak of F 1s at 685.1 eV in Ti_3_C_2_, indicates that some fluoride was left over from the synthesis process [17]. 

The TiO_2_@Ti_3_C_2_ heterostructure exhibits a much stronger O 1s peak intensity when compared to Ti_3_C_2_, indicating the existence of an appreciable amount of oxide in the heterostructure due to the oxidation process and the formation of TiO_2_ [4]. In the spectra results of composite Ag_3_PO_4_/TiO_2_@Ti_3_C_2_, the peaks of P 2p and Ag 3d represent Ag_3_PO_4_ in the composite. Generally, Ag 3d peaks have a higher intensity than Ti 2p, suggesting that many silver elements or their compounds are present on the surface of the solid sample, decorating the TiO_2_@Ti_3_C_2_ flower shape. This result is consistent with the SEM and TEM images discussed previously.

Figure 5 demonstrates the Ti 2p, C 1s, and O 1s high-resolution spectra of Ti_3_C_2_, TiO_2_@Ti_3_C_2_, and Ag_3_PO_4_/TiO_2_@Ti_3_C_2_ (sample A4). The Ti-C peak at 454.26 eV in Ti 2p spectra indicates the presence of Ti_3_C_2_; however, there is no Ti-C peak in TiO_2_@Ti_3_C_2_ [27,28]. Additionally, the intensification of the Ti-O peak (Ti^4+^) at 458.83 eV indicates the formation of TiO_2_ in the flower-shaped TiO_2_@Ti_3_C_2_. The Ti-C bond is absent in TiO_2_@Ti_3_C_2_ in the high-resolution C 1s; however, C–C/C–H, C–O, and O–C = O bonds exist at approximately 284.88, 286.40, and 288.72 eV, respectively [28,29,30]. Owing to the use of TiO_2_@Ti_3_C_2_ as a template for precipitate Ag_3_PO_4_, the Ti 2p and C 1s spectra of Ag_3_PO_4_/TiO_2_@Ti_3_C_2_ (sample A4) are similar to those of TiO_2_@Ti_3_C_2_. 

Nevertheless, the differences in O 1s XPS spectra among the three samples were unavoidable owing to the changing environment in which oxygen was present. Ti_3_C_2_ exhibited a distinct peak attributed to C-Ti-OH linkage at 530.72 and 532.59 eV, whereas the others showed the strongest Ti-O peak at 529.60 eV. Furthermore, Ti_3_C_2_ MXene also exhibited F 1s spectra (Appendix A), which could be ascribed to residual fluorine after the etching steps. The Ag 3d and P 1s spectra of Ag_3_PO_4_/TiO_2_@Ti_3_C_2_ (sample A4) are presented in Appendix A.

The binding energies of 366.83 and 372.91 eV assigned to Ag 3d_5/2_ and Ag 3d_3/2_, respectively, indicate that Ag^+^ is dominated in the composites. The deconvoluted Ag 3d peaks are slightly shifted toward lower binding energy, and the FHMW is also wider. This is attributed to self-corrosion after exposure to the environment for a period of time [22]. The peak of the P 2p spectra located at 131.82 eV corresponds to P^5+^ in the PO_4_^3+^ [31].

### 3.4. Surface Area and Optical Analysis

The N_2_ adsorption–desorption isotherms and the corresponding BET surface areas, pore volumes, and pore sizes of TiO_2_@Ti_3_C_2_, Ag_3_PO_4_/P25, and Ag_3_PO_4_/TiO_2_@Ti_3_C_2_ are shown in Figure 6a and Table 1. Based on the isotherm curves, all samples belong to type IV, and the pore size indicates that they are mesopores [4]. As can be seen, the TiO_2_@Ti_3_C_2_ flowers have the highest BET surface area (53 m^2^g^−1^). After the deposition of Ag_3_PO_4_, the surface area and pore volume of the Ag_3_PO_4_/TiO_2_@Ti_3_C_2_ sample (40 m^2^g^−1^) decreased slightly but remained higher than that of Ag_3_PO_4_/TiO_2_ P25 (27 m^2^g^−1^). This high surface area enables photocatalytic reactions to occur.

The light-harvesting capability is important for evaluating photocatalytic activity as demonstrated by UV-Vis absorption spectra in Figure 6b. Owing to its metallic properties, Ti_3_C_2_ MXene exhibits a nearly horizontal spectrum over the entire wavelength range. The sequential transformations of MXene (metallic material) to the TiO_2_@Ti_3_C_2_ heterostructure (semiconductor material) throughout the processes resulted in the gradual development of the TiO_2_@Ti_3_C_2_ absorption peak [4,16,17]. Its light absorption is improved compared with that of the commercial TiO_2_ P25.

It can be seen that the decoration of Ag_3_PO_4_ on the flower-shaped TiO_2_@Ti_3_C_2_ extends the optical absorption by the combination of Ag_3_PO_4_ and TiO_2_@Ti_3_C_2_. Due to the high surface area of the 3D nanoflower structure, the harvested and scattered light is omnidirectional. In this case, the forward and backward light results in constructive interference, which can extend the photon lifetime and improve the absorbance [16]. These findings further confirm that the Ag_3_PO_4_-deposited TiO_2_@Ti_3_C_2_ composite can enhance the light-harvesting ability, making it a promising photocatalyst under visible illumination.

### 3.5. Photocatalytic Performance

#### 3.5.1. The Effects of Ag_3_PO_4_ Content on the Photodegradation of Rhodamine B

Figure 7a illustrates the photocatalytic degradation of RhB (9 mgL^−1^) using Ag_3_PO_4_, TiO_2_ and Ag_3_PO_4_/TiO_2_@Ti_3_C_2_ composites (samples A1, A2, A3, A4, and A5) at a concentration of 0.5 gL^−1^ under solar light irradiation. Overall, the composite exhibited effective photodegradation reaction under solar light irradiation in a short period of time, with sample A4 displaying significant enhanced photocatalytic performance by demonstrating 97% RhB degradation within 12 min.

As samples A1 and A2 contain less Ag_3_PO_4_ and may be susceptible to self-corrosion and photocorrosion, as discussed in the XRD diffractogram, these samples took 20 min to partially degrade 28.9% and 59.9% RhB, respectively. Although sample A5 contains more Ag_3_PO_4_ than sample A4, there was no significant difference in the photocatalytic activities between samples A5 and A4. When comparing degradation performances within 20 min, pure Ag_3_PO_4_ had a degradation efficiency of 95% and Ag_3_PO_4_/TiO_2_ P25 86%. The lower photocatalytic performance of Ag_3_PO_4_/TiO_2_ P25 is most likely due to self-corrosion in the composite after prolonged storage [22].

Figure 7b shows the calculated rate constant (k) values of 0.017, 0.046, 0.161, 0.289, and 0.278; 0.223; 0.095 min^−1^; and 0.0093 min^−1^ for samples A1–A5, Ag_3_PO_4_, Ag_3_PO_4_/TiO_2_ P25, and flower-like TiO_2_@Ti_3_C_2_, respectively. These results indicate that, among the as-prepared samples, sample A4 exhibited the greatest photocatalysis performance. These results indicate that combining TiO_2_ derived from MXene with Ag_3_PO_4_ can significantly improve its photocatalytic performance when exposed to solar illumination.

#### 3.5.2. Photodegradation of Other Organic Dyes

The three dyes, MB, CV, and MO (10 mg L^−1^, 0.5 gL^−1^) were also photodegraded under solar light irradiation. The adsorption and photocatalytic performance of sample A4 over time for the three organic dyes are shown in Figure 8b and Appendix A. The results indicate that the composite is effective in adsorbing cationic dyes (MB and CV) but has no effect on anionic dye concentration (MO) after 15 min in the dark. It can be attributed to the composite’s surface, which is anionic and negatively charged in the aqueous solution, which enables it to readily attract and adsorb cationic dyes in comparison to its anionic dye counterpart [32]. These results aid in the cationic dyes’ molecules more easily reaching the surface of materials, leading to higher degradation performance in MB and CV compared to MO. After 6 min, the photocatalyst degraded 94.4% of MB, nearly 99.2% CV in 14 min, and 92.4% MO after 40 min. The pseudo-first-order rate constants for MB, CV, and MO were calculated to be 0.489, 0.279, and 0.073 min^−1^, respectively.

Compared with TiO_2_@Ti_3_C_2_, pristine Ag_3_PO_4_, and Ag_3_PO_4_/TiO_2_ P25, the Ag_3_PO_4_/TiO_2_@Ti_3_C_2_ composite exhibited superior photocatalytic activity. Recent publications on photocatalytic materials indicate the possibility of a highly promising photocatalyst for flower shaped TiO_2_@Ti_3_C_2_ with a high surface area. Combining TiO_2_@Ti_3_C_2_ with Ag_3_PO_4_ results in synergetic photodegradation of various dyes. The flower-like structure of TiO_2_ derived from Ti_3_C_2_ can improve its light-harvesting ability, and its high specific surface area allows the solvent to approach the reactive sites more easily.

Additionally, Ag_3_PO_4_ in the composites can shorten the diffusion paths for photoexcited carriers, thereby, inhibiting the recombination of electron–hole pairs [4,25]. The photocatalytic activity of the optimized sample was compared to that of other reported similar photocatalysts. As demonstrated in Table 2, the Ag_3_PO_4_/TiO_2_@Ti_3_C_2_ composite (sample A4) can degrade a variety of pollutants, including both cationic and anionic dyes, in a relatively short period of time, as compared to the state-of-the-art works listed in Table 2.

### 3.6. Scavenger Trapping and Recycling Tests of RhB Degradation

To ascertain the main active species in the photocatalytic reactions, scavenger trapping tests were conducted on RhB photodegradation over the Ag_3_PO_4_/TiO_2_@Ti_3_C_2_ composite (sample A4). The three types of scavengers were tert-butanol (a quencher of •OH), Na_2_-EDTA (a quencher of h^+^), and p-benzoquinone (p-BQ, a quencher of •O^2−^). The dye solution containing a specified amount of catalyst was added to 2 mL of trapping agent (0.01 M), stirred for 15 min in the dark, and exposed to solar light for 20 min. As shown in Figure 9a, the addition of tert-butanol had no effect on the photocatalytic performance, thereby, indicating that free •OH radicals were not the dominant oxidizing species. The presence of Na_2_-EDTA and p-BQ, however, decreased the degradation efficiency to 4.76% and 21.7%, respectively. From these results, it can be concluded that h^+^ and •O_2_^−^ radicals play a critical role in the photodecomposition of RhB.

Along with the photocatalytic efficiency, stability evaluations of photocatalysts are necessary for practical use. Figure 9b shows the RhB degradation efficiency of sample A4 over three consecutive recycling runs. The degradation efficiency was significantly reduced after repeated exposure to solar light, indicating that the photocatalyst was not stable and rapidly deteriorated due to the photocorrosion of Ag_3_PO_4_ NPs [22]. In the first run, a degradation efficiency of 97% was obtained in 12 min, whereas the degradation efficiencies of RhB were approximately 74% and 39% after the second and third runs, respectively. After a few recycling tests under solar light, the photocorrosion of the catalyst resulted in a low photodecomposition rate by forming an uncontrolled amount of Ag^0^, which can agglomerate and hinder the photocatalytic ability [31].

### 3.7. Proposed Photocatalytic Mechanism

Based on the obtained scavenger trapping and recycling results, the photocatalytic mechanism of Ag_3_PO_4_/TiO_2_@Ti_3_C_2_ is schematically shown in Figure 10. Under solar illumination, electrons from the valence band (VB) of Ag_3_PO_4_ could be excited to its conduction band (CB), resulting in the formation of holes in the valence band (VB). Since the E_VB_ of TiO_2_ is more negative than that of Ag_3_PO_4_, holes from Ag_3_PO_4_ also transferred to TiO_2_ and directly degraded RhB, which was absorbed on the surface of TiO_2_ during the oxidization process [25,31]. According to literature, the conduction potential of Ag_3_PO_4_ is higher than the activation energy of single-electron oxygen, preventing photogenerated electrons from being captured by dissolved oxygen. 

Instead, a small amount of Ag is formed as a result of photocorrosion [31]. Electrons and holes in Ag could also be excited by light energy, with electrons migrating to the CB of TiO_2_, while the remaining holes could be recombined with photoexcited electrons in Ag_3_PO_4_ CB, thereby, partially preventing further corrosion [31]. Furthermore, some electrons were transferred from TiO_2_ and Ag_3_PO_4_ to Ti_3_C_2_ MXene, resulting in band bending and the formation of a Schottky junction as well as a uniform Fermi level [4,17,39]. The photo-reduction reaction occurring on the surface of Ti_3_C_2_ and the TiO_2_ conduction band generated •O_2_^−^ radicals, which are required for effective RhB decomposition. The production of •O_2_^−^ and h^+^ could easily degrade a variety of dyes, including RhB, MB, CV, and MO, under solar light. The proposed mechanism is consistent with scavenger trapping experiments and is formulated as follows:**Photo-oxidation reaction:**

Ag_3_PO_4_ + hv → Ag_3_PO_4_ (h^+^ + e^−^)

Ag_3_PO_4_ (h^+^) + TiO_2_ → Ag_3_PO_4_ + TiO_2_ (h^+^)

TiO_2_ (h^+^) + RhB → By-products + CO_2_ + H_2_O

2.
**Photo-reduction reaction:**


Photoexcited electrons and separation of e^−^/h^+^:

Ag^+^ + Ag_3_PO_4_ (e^−^) → Ag + Ag_3_PO_4_

Ag + hv → Ag (h^+^ + e^−^)

Transfer routes and formation of O_2_^●−^:

Ag_3_PO_4_ (e^−^) + Ag (h^+^) → Ag_3_PO_4_ + Ag

Ag (e^−^) + TiO_2_ → Ag + TiO_2_ (e^−^)

TiO_2_ (e^−^) + Ti_3_C_2_ → TiO_2_ + Ti_3_C_2_ (e^−^)

Ag_3_PO_4_ (e^−^) + Ti_3_C_2_ → Ag_3_PO_4_ + Ti_3_C_2_ (e^−^)

Ti_3_C_2_ (e^−^) + O_2_ → Ti_3_C_2_ + O_2_^●−^

TiO_2_ (e^−^) + O_2_ → TiO_2_ + O_2_^●−^

Degradation of pollutants:

O_2_^●−^ + RhB → By-products + CO_2_ + H_2_O

Based on this proposed mechanism, the stability of the composite did not significantly improve. This might be due to a longer electron transfer distance between TiO_2_ and Ti_3_C_2_ than between Ag_3_PO_4_ and Ti_3_C_2_. Second, an unknown amount of Ti_3_C_2_ on the surface and/or insufficient interfacial contact between Ag_3_PO_4_ and Ti_3_C_2_ might lower the composite’s stability. Both reasons could not prevent the rapid formation of Ag, reducing the stability and photocatalytic performance of the composite.

## 4. Conclusions

This study successfully synthesized Ag_3_PO_4_/TiO_2_@Ti_3_C_2_ composites by precipitating Ag_3_PO_4_ NPs on the surface of TiO_2_@Ti_3_C_2_ flowers. Some characterization methods were conducted to investigate the surface morphology, structural composition, surface area, and optical properties of Ag_3_PO_4_/TiO_2_@Ti_3_C_2_. The characteristics of Ag_3_PO_4_-deposited TiO_2_@Ti_3_C_2_ endow superior photocatalytic activities in the comparison with TiO_2_@Ti_3_C_2_, pristine Ag_3_PO_4_, and Ag_3_PO_4_/TiO_2_ P25, particularly sample A4. This composite exhibited excellent photocatalytic performance for various organic dyes (degraded 97% RhB, 94% MB, 99% CV, and 92% MO) within a short period of time when exposed to solar light irradiation. 

The high photocatalytic activity of the composite was due to the high surface area of TiO_2_@Ti_3_C_2_ with small Ag_3_PO_4_ particles (4–10 nm) that can easily reach dye molecules. The e^−^/h^+^ transfer throughout the composite system also contributes to increased charging transfer, expanded light absorption wavelength, and decreased electron–hole pair recombination, which gives synergistic effects for photocatalytic activity. Based on the scavenger trapping and recycling test results, the mechanism was proposed.

## Data Availability

The data presented in this study are available on request from the corresponding author.

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
