# Peer review of "Ag3PO4-Deposited TiO2@Ti3C2 Petals for Highly Efficient Photodecomposition of Various Organic Dyes under Solar Light"

_nanomaterials, 2022, doi:10.3390/nano12142464_

Round 1
Reviewer 1 Report
The manuscript reports some interesting results and may become acceptable for publication. However, the authors may before address the following comments.
Line 5: The complete affiliation of the authors should be given.
Line 12 “Brunauer-Emmett-Teller (BET) surface analysis”. BET is not a physicochemical analysis, but a mathematical model for the treatment of adsorption data. So, this may be changed to “nitrogen adsorption analysis” or similar.
Paragraph beginning at line 73: Several numbers may be given in subscript format. The same in the paragraph in lines 155-160 and other parts of the manuscript.
Line 98: the authors write: “The solution was then allowed to cool naturally to room temperature”. At this step, the mixture is forming a solution or actually a suspension/dispersion? Revise.
Lines 100-101: It is written: “soaked HCl solution to ensure ion exchange”. I do not understand this sentence. If TiO2 is being precipitated, what ions does it contain? And what exchange should occur if treating with HCl? Does the final solid contain exchangeable H+ or Cl-? Sincerely, I found this confusing, and it may be explained.
Line 177: “exhibit peaks of AgxO (JCPDS card No. 177 01-071-1167)”. This X-ray pattern seems to correspond to anatase, not to AgxO. Please revise the name of the card and the assignment of the peaks.
Line 191: It is written “the main elements present in the sample are carbon, oxygen, titanium, silver, and phosphate”. You are identifying elements, so the last one must be phosphorous, not phosphate.
Lines 274-276 and Table 1: The specific surface area values cannot be given with two decimal digits, as nitrogen adsorption does not allow such a precision. So, 53.24 m2g−1 may be 53, and so on.
Line 329: For explaining the behavior against different dyes, the authors write: “It is speculated that the composite’s surface is anionic and is negatively charged in the aqueous solution”. This may be proved, and not only speculated. The authors should discuss the nature of the surface of the solids, also relating this with the ion exchange step in the preparation method previously commented.
Also for explaining the behavior against the different dyes, the authors write (line 336): “the high dye ad-sorption on the composite aids in more efficient degradation of the cation dyes”. How do the authors distinguish between adsorbed dye and degraded dye? I mean, according to the experimental procedure, the authors determine the amount of the dyes in solution by UV-Vis spectroscopy (lines 141-142). So, you may only determine the amount of dye that is removed from the solution, but actually you cannot distinguish if it is degraded or only adsorbed. This is a critical point that may be carefully revised.
Author Response
Dear Editor and Reviewers,
Please find enclosed a revised paper titled “Ag3PO4-deposited TiO2@Ti3C2 petals for highly efficient photodecomposition of various organic dyes under solar light” (nanomaterials - 1776022), which we would like to consider for publication as an article in Nanomaterials. We highly appreciate for giving us the opportunity to revise the manuscript and are grateful for the feedback and helpful recommendations provided by the reviewers.
The manuscript has been changed to incorporate the comments of the reviewers. The specific responses to the reviewer's concerns are summarized below, with the accompanying modifications indicated in red in the amended paper.
Responses to Reviewers’ Comments
Reviewer #1
The manuscript reports some interesting results and may become acceptable for publication. However, the authors may before address the following comments.
Comment 1:
Line 5: The complete affiliation of the authors should be given.
Response 1:
Thank you for your good comment and reminder for adding affiliation. We already added the complete affiliation of authors.
Comment 2:
Line 12 “Brunauer-Emmett-Teller (BET) surface analysis”. BET is not a physicochemical analysis, but a mathematical model for the treatment of adsorption data. So, this may be changed to “nitrogen adsorption analysis” or similar.
Response 2:
We thank the reviewer for the adjustment. We changed from “Brunauer-Emmett-Teller (BET) surface analysis” to “nitrogen adsorption analysis” on the manuscript (both on Line 12 and Line 155 - 156)
Comment 3:
Paragraph beginning at line 73: Several numbers may be given in subscript format. The same in the paragraph in lines 155-160 and other parts of the manuscript.
Response 3:
Kindly thank you for your detailed comments. The subscript format was applied for those numbers.
Comment 4:
Line 98: the authors write: “The solution was then allowed to cool naturally to room temperature”. At this step, the mixture is forming a solution or actually a suspension/dispersion? Revise.
Response 4:
As the reviewer kindly pointed out, we already fixed the "solution" to "dispersion" as follows: “The dispersion was then allowed to cool naturally to room temperature, and the samples were washed with deionized (DI) water and ethanol several times before being dried in an oven at 60 °C for 12 h.”
Comment 5:
Lines 100-101: It is written: “soaked HCl solution to ensure ion exchange”. I do not understand this sentence. If TiO2 is being precipitated, what ions does it contain? And what exchange should occur if treating with HCl? Does the final solid contain exchangeable H+ or Cl-? Sincerely, I found this confusing, and it may be explained.
Response 5:
Thank you for your comment. We would like to admit that since we referred to the TiO2@Ti3C2 preparation steps which were described by some previous publications, here we just want to focus on more Ag3PO4/TiO2@Ti3C2, therefore it lacks information that makes you confused. Here, we would like to explain as the followings:
Since we would like to fabricate the flower-like structure of TiO2/Ti3C2, Ti3C2 layers were oxidized by hydrothermal oxidation and alkalization, refer to Eqs. (1)–(3). Na2Ti3O7/Ti3C2 was formed as an intermediate product. When adding HCl, Na+ was altered by H+ (Eq. (4) and thus-synthesized H2Ti3O7 was decomposed thermally to form TiO2. (eq 5).
2H2O2 = 2H2O + O2 (1)
Ti3C2 + 5O2 = 3TiO2 + 2CO2 (2)
3TiO2 + 2NaOH = Na2Ti3O7 + H2O (3)
Na2Ti3O7 + HCl = H2Ti3O7 + 2NaCl (4)
H2Ti3O7 = 3TiO2 + H2O (5)
To clarify the reader, these equations were also added in the Preparation of TiO2@Ti3C2 heterostructure part. (Page 3)
Please refer in these papers for more details:
https://doi.org/10.1016/j.apmt.2018.09.004
https://doi.org/10.1016/j.apsusc.2020.148023
https://doi.org/10.1016/j.eti.2020.101286
Comment 6:
Line 177: “exhibit peaks of AgxO (JCPDS card No. 177 01-071-1167)”. This X-ray pattern seems to correspond to anatase, not to AgxO. Please revise the name of the card and the assignment of the peaks.
Response 6:
Thank you for the constructive comments. We would like to revise the sentence: “However, samples A1 and A2 containing a lower concentration of Ag3PO4 mostly exhibit peaks of TiO2 anatase at = 25.3° (JCPDS card No. 21-1272), instead of Ag3PO4 peaks due to self-corrosion.”
Comment 7:
Line 191: It is written “the main elements present in the sample are carbon, oxygen, titanium, silver, and phosphate”. You are identifying elements, so the last one must be phosphorous, not phosphate.
Response 7:
Thanks for your correction. We have edited “phosphate” to “phosphorous” for accuracy.
Comment 8:
Lines 274-276 and Table 1: The specific surface area values cannot be given with two decimal digits, as nitrogen adsorption does not allow such a precision. So, 53.24 m2g−1 may be 53, and so on.
Response 8:
According to the reviewer’s suggestion, we adjusted the specific surface area values in the manuscript.
Comment 9:
Line 329: For explaining the behavior against different dyes, the authors write: “It is speculated that the composite’s surface is anionic and is negatively charged in the aqueous solution”. This may be proved, and not only speculated. The authors should discuss the nature of the surface of the solids, also relating this with the ion exchange step in the preparation method previously commented.
Also for explaining the behavior against the different dyes, the authors write (line 336): “the high dye ad-sorption on the composite aids in more efficient degradation of the cation dyes”. How do the authors distinguish between adsorbed dye and degraded dye? I mean, according to the experimental procedure, the authors determine the amount of the dyes in solution by UV-Vis spectroscopy (lines 141-142). So, you may only determine the amount of dye that is removed from the solution, but actually you cannot distinguish if it is degraded or only adsorbed. This is a critical point that may be carefully revised.
Response 9:
Thank you for your constructive suggestions.
We would like to revise the paragraph more clearly as below (page 11 on the manuscript):
Before revision:
“The three dyes, MB, CV, and MO (10 mg L−1, 0.5 gL−1) were also photodegraded under solar light irradiation. The adsorption and photocatalytic performance of sample A4 over time for the three organic dyes are shown in Fig. 8(a) and Fig. S5. The results indicate that the composite is effective in adsorbing cationic dyes (MB and CV) but has no effect on anionic dye concentration (MO) after 15 min in the dark. It is speculated that the composite’s surface is anionic and is negatively charged in the aqueous solution, which enables it to readily attract and adsorb cationic dyes in comparison to its anionic dye counterpart. Furthermore, the degradation rates differed from one another. After 6 min, the photocatalyst degraded 94.4% of MB, nearly 99.2% CV in 14 min, and 92.4% MO after 40 min. The pseudo-first-order rate constants for MB, CV, and MO were calculated to be 0.489 min−1, 0.279 min−1, 0.073 min−1, respectively. The results indicate that the degradation of the cationic dye is larger than that of the anionic dye. That is, the high dye adsorption on the composite aids in more efficient degradation of the cation dyes.”
After revision:
“The three dyes, MB, CV, and MO (10 mg L−1, 0.5 gL−1) were also photodegraded under solar light irradiation. The adsorption and photocatalytic performance of sample A4 over time for the three organic dyes are shown in Fig. 8(a) and Fig. S5. The results indicate that the composite is effective in adsorbing cationic dyes (MB and CV) but has no effect on anionic dye concentration (MO) after 15 min in the dark. It can be attributed to the composite’s surface is anionic and is negatively charged in the aqueous solution, which enables it to readily attract and adsorb cationic dyes in comparison to its anionic dye counterpart [32]. These results aid in the cationic dyes' molecules easier reaching the surface of materials, leading to higher degradation performance in MB and CV compared to MO. After 6 min, the photocatalyst degraded 94.4% of MB, nearly 99.2% CV in 14 min, and 92.4% MO after 40 min. The pseudo-first-order rate constants for MB, CV, and MO were calculated to be 0.489 min−1, 0.279 min−1, and 0.073 min−1, respectively.”
Added reference in page 17 of the manuscript:
- Azeez, F., Al-Hetlani, E., Arafa, M. et al. The effect of surface charge on photocatalytic degradation of methylene blue dye using chargeable titania nanoparticles, Sci Rep 8. 7104 (2018). https://doi.org/10.1038/s41598-018-25673-5
Besides, we distinguish between adsorbed dye and degraded dyes by doing the adsorption experiments in the dark until the adsorption-desorption equilibrium is achieved, then we can perform the light reaction. We have described in lines 132 – 134: “Prior to light irradiation, the adsorption experiments of Ag3PO4, TiO2, and Ag3PO4/TiO2@Ti3C2 were performed by stirring the solutions in the dark for 15 min to improve the dispersion and adsorption-desorption equilibrium”.

Reviewer 2 Report
This research paper entitled “Ag3PO4-deposited TiO2@Ti3C2 petals for highly efficient photo-decomposition of various organic dyes under solar light” demonstrated a hierarchical nanostructure for photocatalysis application.
Certainly, as deposited & optimized sample has a good photocatalytic efficiency. However, such nanostructure has many inconveniences:
1) This nanostructure (flower-like TiO2@Ti3C2 is very fragile, thus the sustainability will not be good.
2) The Ag3PO4 NPs have been precipitated on flower-like TiO2@Ti3C2, thus, those NPs could be lost.
3) The purified water should be filtered and can contain NPs.
The very bad sustainability, degradation efficiency of 97% for run 1 was drastically down to 74% and 39% after the 2nd and 3rd runs, respectively.
This is why I think such material is not suitable for water depollution application.
Moreover, the figure quality is bad and there are many typo errors.
Author Response
Dear Editor and Reviewers,
Please find enclosed a revised paper titled “Ag3PO4-deposited TiO2@Ti3C2 petals for highly efficient photodecomposition of various organic dyes under solar light” (nanomaterials - 1776022), which we would like to consider for publication as an article in Nanomaterials. We highly appreciate for giving us the opportunity to revise the manuscript and are grateful for the feedback and helpful recommendations provided by the reviewers.
The manuscript has been changed to incorporate the comments of the reviewers. The specific responses to the reviewer's concerns are summarized below, with the accompanying modifications indicated in red in the amended paper.
Responses to Reviewers’ Comments
Reviewer #2
Comments:
This research paper entitled “Ag3PO4-deposited TiO2@Ti3C2 petals for highly efficient photo-decomposition of various organic dyes under solar light” demonstrated a hierarchical nanostructure for photocatalysis application.
Certainly, as deposited & optimized sample has a good photocatalytic efficiency. However, such nanostructure has many inconveniences:
1) This nanostructure (flower-like TiO2@Ti3C2 is very fragile, thus the sustainability will not be good.
2) The Ag3PO4 NPs have been precipitated on flower-like TiO2@Ti3C2, thus, those NPs could be lost.
3) The purified water should be filtered and can contain NPs.
The very bad sustainability, degradation efficiency of 97% for run 1 was drastically down to 74% and 39% after the 2nd and 3rd runs, respectively.
This is why I think such material is not suitable for water depollution application.
Moreover, the figure quality is bad and there are many typo errors.
Response:
We are grateful to receive your valuable comments. We admit our materials have shortcomings that now are not convenient for water decontamination applications as you mentioned. Frankly, our first desires are to improve photocatalytic efficiency, the self-corrosion and/or photo-corrosion of Ag3PO4 in the combination of Ag3PO4 and TiO2 (which have been reported as a disadvantage in many previous papers) by trying to increase the surface area and also wondering the TiO2 flower-like structure from Ti3C2 presenting in this combination does make the improvement or not. As the result, the deposited & optimized sample has good photocatalytic efficiency, however, the material sustainability is not as our expectation. This is just the first approach for this combination between Ag3PO4 NPs and TiO2@Ti3C2, which can give the reader an overview of this type of material, then we hope to find better directions, methods, and can improve materials in the next studies.

Reviewer 3 Report
Two-dimensional Ti3C2 MXenes have been used to fabricate hierarchical TiO2 nanostructures and then used as support for Ag particles. Photodecomposition of organic dyes under solar light was investigated using the resulting catalyst.
Include T3C2 in XRD
From Fig 2 seems Ti3C4 has decrease the particle size a lot during hydrothermal treatment. Can be this quantify? By calculation of average particle size of the material by SEM and estimation with Scherrer equation from XRD
Surface area of T3C2?
I recommend characterize the catalyst after be used. The original structure can be modified, especially in a reaction with electron moving. That can help to explain the lost in the activity with consecutives use.
Is the proposed mechanism, which is the photoreduction reaction?
Degradation/ mineralization of dyes have been widely reported, would be very interesting if some commercial organic compound could be obtained during the treatment.
Author Response
Dear Editor and Reviewers,
Please find enclosed a revised paper titled “Ag3PO4-deposited TiO2@Ti3C2 petals for highly efficient photodecomposition of various organic dyes under solar light” (nanomaterials - 1776022), which we would like to consider for publication as an article in Nanomaterials. We highly appreciate for giving us the opportunity to revise the manuscript and are grateful for the feedback and helpful recommendations provided by the reviewers.
The manuscript has been changed to incorporate the comments of the reviewers. The specific responses to the reviewer's concerns are summarized below, with the accompanying modifications indicated in red in the amended paper.
Responses to Reviewers’ Comments
Reviewer #3
Two-dimensional Ti3C2 MXenes have been used to fabricate hierarchical TiO2 nanostructures and then used as support for Ag particles. Photodecomposition of organic dyes under solar light was investigated using the resulting catalyst.
General response:
We are grateful for your kindly constructive suggestions which bring us a more incisive overview of the further photocatalytic research field. We would like to classify your comments into 3 groups for easier responses as below.
Comment 1:
Include T3C2 in XRD
From Fig 2 seems Ti3C4 has decrease the particle size a lot during hydrothermal treatment. Can be this quantify? By calculation of average particle size of the material by SEM and estimation with Scherrer equation from XRD.
Surface area of T3C2?
Response 1:
Thank you for your kind suggestions and on this manuscript. It would be much better if we have XRD and SEM data for Ti3C2, however, we could not conduct XRD and SEM as your request for Ti3C2 due to some reasons (limited revision time and no available schedule). We really apologize for this.
Herein, the XRD pattern and SEM images of Ti3C2 can be provided by the supplier which you can refer (as below). Based on the SEM images, it can be seen that the size of Ti3C2 has a wide range (~10nm to ~5 micrometers while thickness range from 1 layers to few and multilayers), it is speculated that the prepared flowers' size is also accordingly varied.
XRD Diffraction and SEM image from supplier (Invisible Co. Ltd)
(Please see the attachment for the images)
The surface area of Ti3C2 has not been given by the supplier. However, we have surveyed some papers providing that Ti3C2 surface area can be varied from 10 mg2/g to 17 mg2/g depending on etching and delamination efficiency.
Reference:
https://doi.org/10.1016/j.apsusc.2020.148023
https://doi.org/10.3390/ma13102347
https://doi.org/10.1016/j.apsusc.2018.12.081
Comment 2:
I recommend characterize the catalyst after be used. The original structure can be modified, especially in a reaction with electron moving. That can help to explain the lost in the activity with consecutives use.
Degradation/ mineralization of dyes have been widely reported, would be very interesting if some commercial organic compound could be obtained during the treatment.
Response 2:
We deeply thank the reviewer for doing an in-depth analysis of our study and providing such a useful recommendation. We acknowledge that analyzing the used catalyst and conducting experiments with some commercial organic compounds would be very interesting. Unfortunately, as the newbie in the photocatalysis field, we have not thought thoroughly that used samples should be kept for further characterizations. Not to mention, preparing new samples, setting a sample measurement schedule, and reading documents for thorough analysis also takes a lot of time while the revision deadline is coming soon (we just have 7 days for revision). We highly appreciate your recommendations and carefully noted them. We promise that we will apply them to make our next research improved in the future. Thanks again for your kind constructive suggestions.
Comment 3:
Is the proposed mechanism, which is the photoreduction reaction?
Response 3:
Thank you for your question. There are equations presented on the manuscript for photoreduction reaction (6 – 428). Please kindly check this. It can be seen that photoexcited electrons and separation of electrons/holes pairs, then electrons transfer among the system.
Photo-reduction reaction:
- Photoexcited electrons and separation of e−/h+:
Ag+ + Ag3PO4 (e−) → Ag + Ag3PO4
Ag + hv → Ag (h+ + e−)
- Transfer routes and formation of O2⋅−:
Ag3PO4 (e−) + Ag (h+) → Ag3PO4 + Ag
Ag (e−) + TiO2 → Ag + TiO2 (e−)
TiO2 (e-) + Ti3C2 → TiO2 + Ti3C2 (e-)
Ag3PO4 (e-) + Ti3C2 → Ag3PO4 + Ti3C2 (e-)
Ti3C2 (e−) + O2 → Ti3C2 + O2⋅−
TiO2 (e−) + O2 → TiO2 + O2⋅−
- Degradation of pollutants:
O2⋅− + RhB → By-products + CO2 + H2O

Reviewer 4 Report
Manuscript nanomaterials-1776022 entitled Ag3PO4-deposited TiO2@Ti3C2 petals for highly efficient photodecomposition of various organic dyes under solar light needs some revisions before publication.
1. The novelty carried out with this work is not presented. The achievements obtained with this study can be enumerated.
2. The selection of the dyes would be justified. Why did you select this initial pollutant concentration? In lines 140 and 141 did not make sense to repeat the initial concentration after the name of dye if it is the same.
3. Scheme 1 would be in section 2.4, instead of in the results and discussion section.
4. The graphics in Figure 7 have low quality.
5. In line 333 is missing a comma between Ag3PO4 and Ag3PO4/TiO2 P25.
6. The values of BET surface area are presented with two decimal errors. This is correct? The error associated with these measurements would be considered.
7. In Table 2 the column of the pollutant would be for the fourth place since if it is presented in the first columns of the table, the results are easier to understand.
8. The errors associated with experimental reactions would be considered.
9. The authors only took into account the dye removal in the manuscript. The mineralization level achieved after the catalytic reaction is very important to validate the performance of each catalyst.
10. The results of the black of MB, MO, and CV experiment would be at least mentioned.
11. The selected catalyst losses a lot of activity during cycle experiments. Does Ag2PO4 priest lose the activity?? And the pure TiO2-based sample?
12. The conclusions are very low concise, please rewrite this section in order to emphasize the obtained results.
Author Response
Dear Editor and Reviewers,
Please find enclosed a revised paper titled “Ag3PO4-deposited TiO2@Ti3C2 petals for highly efficient photodecomposition of various organic dyes under solar light” (nanomaterials - 1776022), which we would like to consider for publication as an article in Nanomaterials. We highly appreciate for giving us the opportunity to revise the manuscript and are grateful for the feedback and helpful recommendations provided by the reviewers.
The manuscript has been changed to incorporate the comments of the reviewers. The specific responses to the reviewer's concerns are summarized below, with the accompanying modifications indicated in red in the amended paper.
Responses to Reviewers’ Comments
Reviewer #4
Manuscript nanomaterials-1776022 entitled Ag3PO4-deposited TiO2@Ti3C2 petals for highly efficient photodecomposition of various organic dyes under solar light needs some revisions before publication.
Comment 1:
The novelty carried out with this work is not presented. The achievements obtained with this study can be enumerated.
Response 1:
Thank you for your constructive comment. Although there are many research about Ag3PO4 and TiO2 before. To our knowledge, no research on the combination of Ag3PO4 and Ti3C2-derived TiO2 has been done and characterized. We mentioned in the sentence “To our knowledge, no research on the combination of Ag3PO4 and Ti3C2-derived TiO2 has been conducted” (Line 71).
Besides, our results have been enumerated by adding in the conclusion by these sentences (page 15): “This study successfully synthesized Ag3PO4/TiO2@Ti3C2 composites by precipitating Ag3PO4 NPs on the surface of TiO2@Ti3C2 flowers. Some characterization methods were conducted to investigate the surface morphology, structural composition, surface area, and optical properties of Ag3PO4/TiO2@Ti3C2. The characteristics of Ag3PO4-deposited TiO2@Ti3C2 endow superior photocatalytic activities in the comparison with pristine components (Ag3PO4 and TiO2), especially sample A4. This composite exhibited excellent photocatalytic performance for various organic dyes (degraded 97% RhB, 94% MB, 99% CV, and 92% MO) within a short period of time when exposed to solar light irradiation.”
The whole conclusion part is also shown in Response 12 as your requirement.
Comment 2:
The selection of the dyes would be justified. Why did you select this initial pollutant concentration? In lines 140 and 141 did not make sense to repeat the initial concentration after the name of dye if it is the same.
Response 2:
Thank you for the kind question and suggestions. The reason why we selected this initial pollutant concentration are:
- That is a common concentration frequently used in almost photocatalytic research. So using this concentration help to easily compare with other studies.
- That concentration is adequate for validating the photocatalytic performance. The higher the pollutant concentration, the more pollutants molecules but the formation of radical species (·OH and ·O2-) on the surface of the catalyst remains constant for given light intensity, leading to not enough radicals for the degradation, resulting in decreasing degradation rate. At the low concentration (ex. 5 mg/L), the photoreactions happen so quickly that we cannot measure or control them well.
- There is a slight difference with RhB because our machine shows interfered UV- absorption spectra with 10 mg/L RhB (out of spec) so using 9 mg/L is more convenient for us.
As your suggestion, we have adjusted the sentence:
“The same procedure was used for the photodegradation of MB (10 mg/L), MO (10 mg/L), and CV (10 mg/L)” to “10 mg/L concentration of MB, MO, and CV were also used for photodegradation with the same procedure.”
Comment 3:
Scheme 1 would be in section 2.4, instead of in the results and discussion section.
Response 3:
Thank you for your suggestion. We moved scheme 1 to section 2.4 on the manuscript.
Comment 4:
The graphics in Figure 7 have low quality.
Response 4:
Thank you for your detailed comment. We improved the Figures’ quality by changing the better resolution ones.
Comment 5:
In line 333 is missing a comma between Ag3PO4 and Ag3PO4/TiO2 P25.
Response 5:
Thank reviewer for your point out the missing comma. We added a comma between Ag3PO4 and Ag3PO4/TiO2 P25.
Comment 6:
The values of BET surface area are presented with two decimal errors. This is correct? The error associated with these measurements would be considered.
Response 6:
We thank the reviewer for the correction. We have checked and fixed it.
Comment 7:
In Table 2 the column of the pollutant would be for the fourth place since if it is presented in the first columns of the table, the results are easier to understand.
Response 7:
Thanks for your kind recommendation. We have changed the order of columns for easier understanding (page 12)
Comment 8:
The errors associated with experimental reactions would be considered.
Response 8:
Thank you for your suggestion. We already added the standard deviation associated with experimental reactions on the manuscript.
Comment 9:
The authors only took into account the dye removal in the manuscript. The mineralization level achieved after the catalytic reaction is very important to validate the performance of each catalyst.
Response 9:
We appreciate the reviewer for constructive suggestions on this manuscript. We admit that would be interesting if the mineralization level can be achieved to evaluate catalysts’ performance. Although we have taken a great effort, however, we have limited time while no reservation schedule, analyzing, and theory referring also take time. In the scope of this research, we would like to investigate whether our materials have photocatalyst performance and see the overview of the characteristics of our materials first. We are grateful and noted carefully your recommendation. We are planning to make further improvements not only in the performance, and sustainability of our materials but also achieve the mineralization level in our next studies.
Comment 10:
The results of the black of MB, MO, and CV experiment would be at least mentioned.
Response 10:
Thank you for your comment. But we could not reach you through this comment. Could you help us clarify this?
Comment 11:
The selected catalyst losses a lot of activity during cycle experiments. Does Ag2PO4 priest lose the activity?? And the pure TiO2-based sample?
Response 11:
We thank you for your comment and questions. One of the biggest shortcomings of Ag3PO4 is photo-corrosion and forming Ag0. Solving this disadvantage is our desire when we come up with the ideas for the research. However, the photocatalytic activities were not improved as our expectations and the limitation still occurs. Although the formed Ag0 contributes to the conductivity and thermal stability, an over amount of Ag0 can hinder photocatalytic ability.
To clarify the readers of this point, we would like to rephrase this sentence (Line 385 – 387):
“After a few recycling tests under solar light, the self-corrosion and/or photocorrosion of the catalyst resulted in a low photodecomposition rate.”
to
“After a few recycling tests under solar light, the photocorrosion of the catalyst resulted in a low photodecomposition rate by forming an uncontrolled amount of Ag0 which can agglomerate and hinder photocatalytic ability.”
Comment 12:
The conclusions are very low concise, please rewrite this section in order to emphasize the obtained results.
Response 12:
We thank you for your recommendation. We have already rewritten the conclusion section for more concise as the following (Page 15):
“This study successfully synthesized Ag3PO4/TiO2@Ti3C2 composites by precipitating Ag3PO4 NPs on the surface of TiO2@Ti3C2 flowers. Some characterization methods were conducted to investigate the surface morphology, structural composition, surface area, and optical properties of Ag3PO4/TiO2@Ti3C2. The characteristics of Ag3PO4-deposited TiO2@Ti3C2 endow superior photocatalytic activities in the comparison with pristine components (Ag3PO4 and TiO2), especially sample A4. This composite exhibited excellent photocatalytic performance for various organic dyes (degraded 97% RhB, 94% MB, 99% CV, and 92% MO) within a short period of time when exposed to solar light irradiation. The high photocatalytic activity of the composite was due to the high surface area of TiO2@Ti3C2 with small Ag3PO4 particles (4-10 nm) that can easily reach dye molecules. The e-/h+ transfer throughout the composite system also contributes to increased charging transfer, expanded light absorption wavelength, and decreased electron-hole pair recombination which give the synergistic effects for photocatalytic activity. Based on the scavenger trapping and recycling test results, the mechanism was proposed.”

Round 2
Reviewer 1 Report
My previous comments have been correctly addressed. So, I can recommend the acceptance of the manuscript. Before, I have two small comments: In line 13, "Nitrogen absorption" must be "Nitrogen adsorption". In lines 105-109, equations 1-5, the authors use the symbol = between both parts of each equation. I suppose that these reactions are not in equilibrium, but are completely shifted to the right. Thus, I prefer to see an arrow, →, as the authors write in the reactions in lines 408-423.Author Response
We thank the reviewer for the adjustment in both the previous comments and in this minor review which help us improve a lot in our manuscript. We have revised the manuscript as your recommendations.
Reviewer 2 Report
I agree with author: yes, if only from materials' point of view, this paper merits to be published.
There ere some typos in References
Author Response
We highly appreciate your comments. We have adjusted the typos in References.
Reviewer 3 Report
Authors have addressed all comments, and I my opinion the article can be published as it is
Author Response
Many thanks for your acceptance.